# Interface between 40S exit channel protein uS7/Rps5 and eIF2α modulates start codon recognition in vivo

Jyothsna Visweswaraiah, Alan G Hinnebusch*

Laboratory of Gene Regulation and Development, Eunice Kennedy Shriver National Institute of Child Health and Human Development, National Institutes of Health, Bethesda, United States

**Abstract** The eukaryotic pre-initiation complex (PIC) bearing the eIF2·GTP·Met-tRNA$_i^{Met}$ ternary complex (TC) scans the mRNA for an AUG codon in favorable context. AUG recognition evokes rearrangement of the PIC from an open, scanning to a closed, arrested conformation. Cryo-EM reconstructions of yeast PICs suggest remodeling of the interface between 40S protein Rps5/uS7 and eIF2α between open and closed states; however, its importance was unknown. uS7 substitutions disrupting eIF2α contacts favored in the open complex increase initiation at suboptimal sites, and uS7-S223D stabilizes TC binding to PICs reconstituted with a UUG start codon, indicating inappropriate rearrangement to the closed state. Conversely, uS7-D215 substitutions, perturbing uS7-eIF2α interaction in the closed state, confer the opposite phenotypes of hyperaccuracy and (for D215L) accelerated TC dissociation from reconstituted PICs. Thus, remodeling of the uS7/eIF2α interface appears to stabilize first the open, and then the closed state of the PIC to promote accurate AUG selection in vivo.

*For correspondence:
ahinnebusch@nih.gov

## Introduction

Accurate identification of the translation initiation codon is critical to ensure synthesis of the correct cellular proteins. In eukaryotic cells this process generally occurs by a scanning mechanism, wherein the small (40S) ribosomal subunit first recruits Met-tRNA$_i$ in a ternary complex (TC) with eIF2-GTP in a reaction stimulated by eIFs 1, 1A, and 3. The resulting 43S pre-initiation complex (PIC) attaches to the mRNA 5' end and scans the 5'UTR for an AUG with favorable surrounding sequence, particularly at the −3 and +4 positions, to identify the correct start codon and assemble a 48S PIC. In the scanning PIC, Met-tRNA$_i$ is not tightly bound to the peptidyl (P) site of the 40S subunit, and this relatively unstable 'P$_{OUT}$' state is thought to facilitate sampling of successive triplets entering the P site for complementarity to the anticodon of Met-tRNA$_i$. The GTP bound to eIF2 in the TC can be hydrolyzed, dependent on GTPase activating protein eIF5, but P$_i$ release is blocked by eIF1, which also impedes full accommodation of Met-tRNA$_i$ in the P site. Start codon recognition triggers dissociation of eIF1 from the 40S subunit, which gates P$_i$ release from eIF2-GDP·P$_i$ and permits highly stable binding of Met-tRNA$_i$ in the 'P$_{IN}$' state. Interaction of the eIF1A NTT with the codon:anticodon duplex helps to stabilize the closed, P$_{IN}$ state (**Figure 1**). Subsequent dissociation of eIF2-GDP and other eIFs from the 48S PIC enables eIF5B-catalyzed subunit joining and formation of an 80S initiation complex with Met-tRNA$_i$ base-paired to AUG in the P site (reviewed in *Hinnebusch (2014)*).

A recent cryo-EM structure of a reconstituted partial yeast 48S PIC (py48S) with Met-tRNA$_i$ bound in the P$_{IN}$ state revealed extensive interactions between Met-tRNA$_i$ and all three domains of the α-subunit of eIF2 within the TC. The eIF2α occupies the exit (E) decoding site, adjacent to the P site, with eIF2α domain-1 mimicking the anticodon stem-loop (ASL) of an E site-bound tRNA and

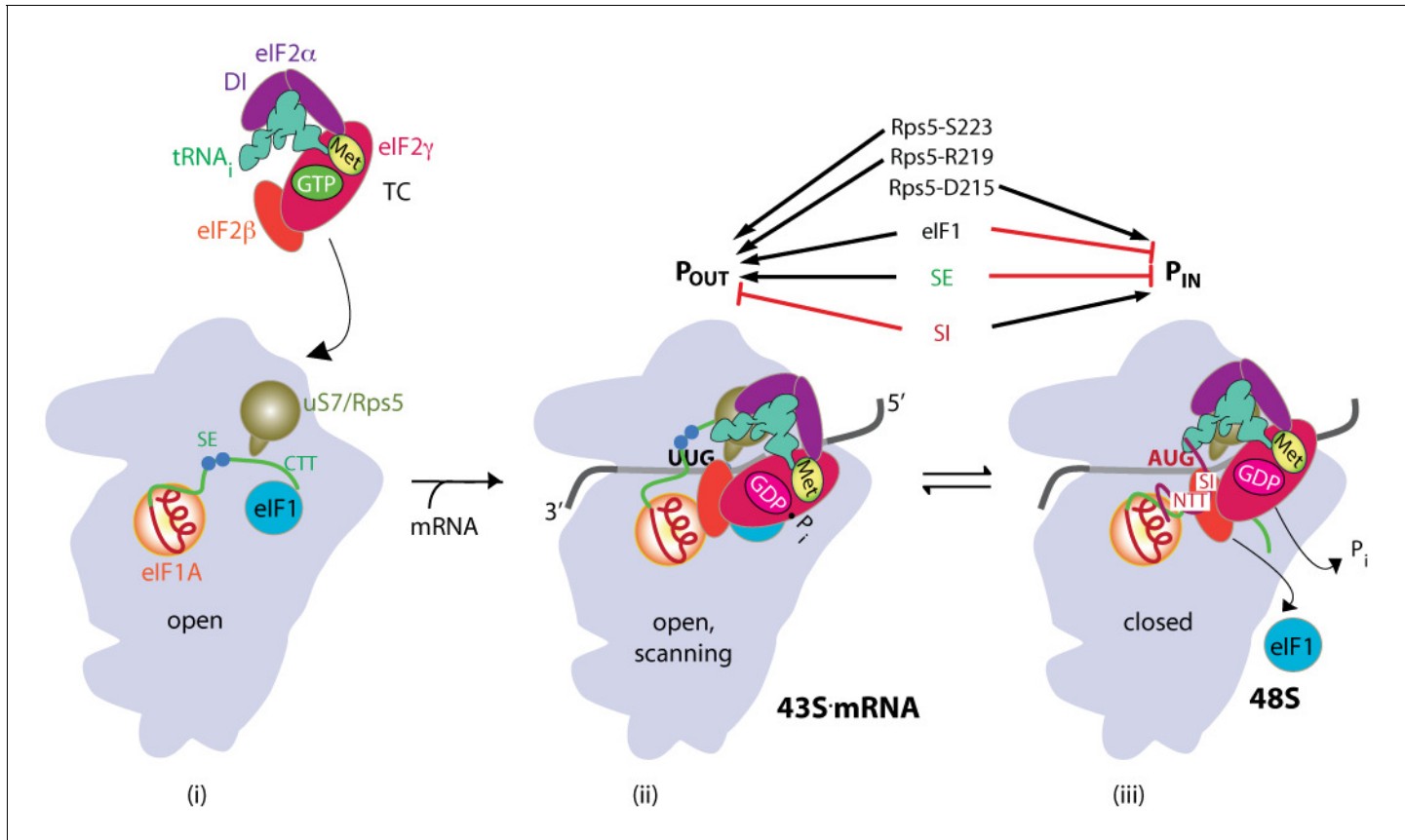

**Figure 1.** Model describing conformational rearrangements of the PIC during scanning and start codon recognition. (i) eIF1 and the scanning enhancers (SEs) in the CTT of eIF1A stabilize an open conformation of the 40S subunit to which TC rapidly binds. uS7 is located in the mRNA exit channel of the 40S; (ii) The 43S PIC in the open conformation scans the mRNA for the start codon with Met-tRNA$_i$ bound in the $P_{OUT}$ state and uS7 interacting with eIF2α-D1. eIF2 can hydrolyze GTP to GDP•P$_i$, but release of P$_i$ is blocked. (iii) On AUG recognition, Met-tRNA$_i$ moves from the $P_{OUT}$ to $P_{IN}$ state, clashing with eIF1 and the CTT of eIF1A, provoking displacement of the eIF1A CTT from the P site, dissociation of eIF1 from the 40S subunit, and P$_i$ release from eIF2. The NTT of eIF1A, harboring scanning inhibitor (SI) elements, adopts a defined conformation and interacts with the codon: anticodon helix. The eIF2α-D1/uS7 interface is remodeled. (Above) Arrows summarize that eIF1 and the eIF1A SE elements promote $P_{OUT}$ and impede transition to $P_{IN}$ state, whereas the scanning inhibitor (SI) element in the NTT of eIF1A stabilizes the $P_{IN}$ state. Results presented below indicate that uS7/Rps5 residue D215 promotes the closed conformation, whereas R219 and S223 enhance the open state (Adapted from *Hinnebusch, 2014*).

contacting the −2 and −3 'context' nucleotides in mRNA just upstream of the AUG codon (*Figure 2A–B*). eIF2α-D1 also interacts with the C-terminal helix of 40S ribosomal protein uS7 (Rps5 in yeast), whose β-hairpin projects into the mRNA exit channel and additionally interacts with the −3 mRNA nucleotide (*Hussain et al., 2014*) (*Figure 2A–C*). Proximity of eIF2α-D1 and the uS7 hairpin with the −3 nucleotide was also observed in structures of partial mammalian 43S (*Hashem et al., 2013*) and 48S PICs (*Lomakin and Steitz, 2013*) and detected in cross-linking analyses of reconstituted mammalian PICs (*Pisarev et al., 2006*; *Sharifulin et al., 2013*); and there is biochemical evidence that recognition of the AUG context nucleotides requires eIF2α (*Pisarev et al., 2006*).

Mutations have been identified in yeast initiation factors, including eIF1, eIF5, and the three subunits of eIF2, that reduce initiation accuracy and increase utilization of near-cognate triplets, particularly UUG, in place of AUG as start codons, conferring the Sui⁻ (Suppressor of initiation codon) phenotype (*Donahue, 2000*). Previously, we showed that substitutions of several residues in the β-hairpin of uS7 suppress the elevated UUG initiation conferred by Sui⁻ variants of eIF2β (*SUI3–2*) or eIF5 (*SUI5*), displaying the Ssu⁻ (Suppressor of Sui⁻) phenotype. Consistent with this, one such Ssu⁻ substitution in the hairpin loop (R148E, *Figure 2B*) was found to destabilize TC binding to reconstituted 48S PICs containing a UUG start codon in the mRNA. Substitutions of Glu-144 in β-strand 1 of the hairpin, or the nearby residue Arg-225 at the C-terminus of uS7 (*Figure 2B*), also reduced

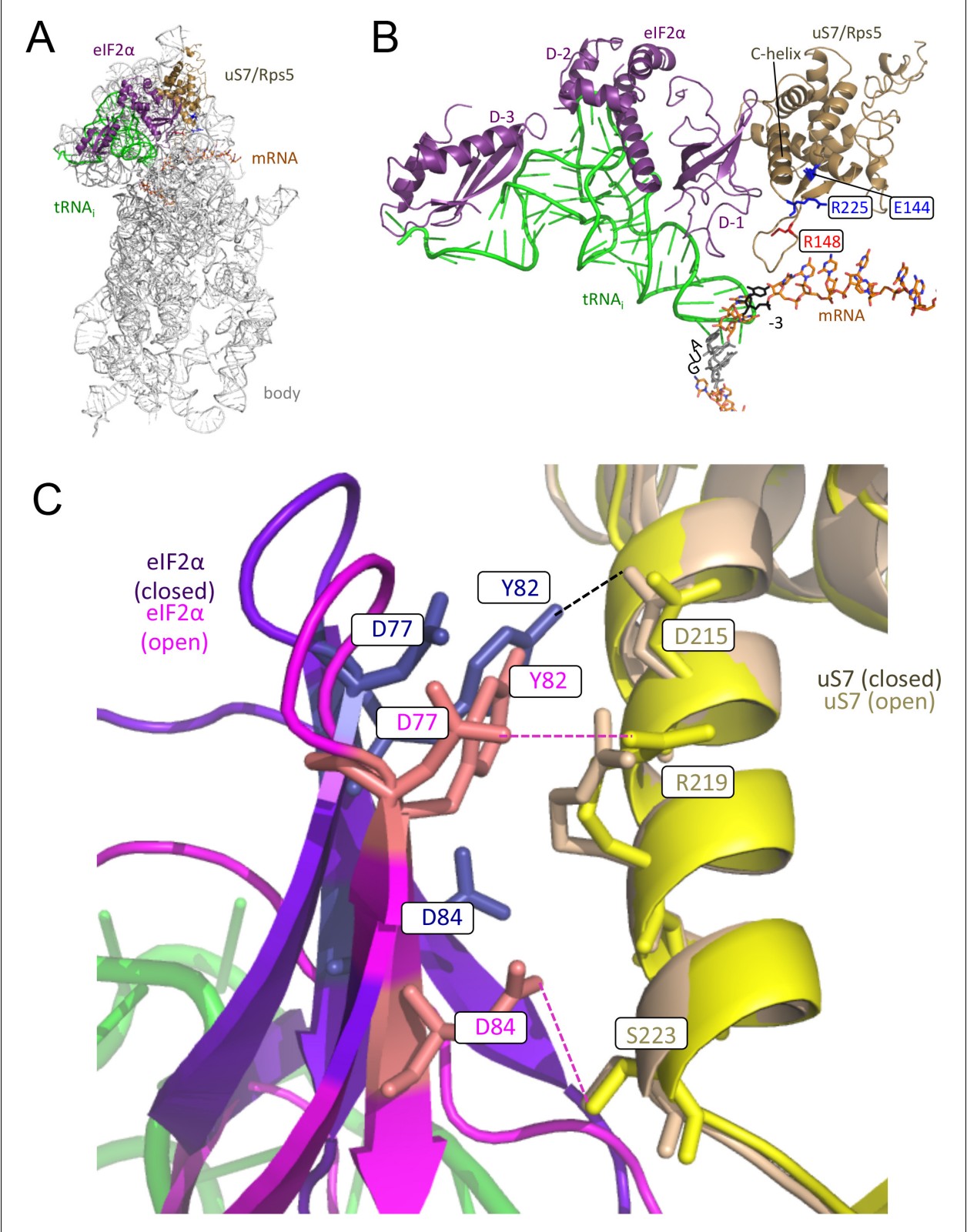

**Figure 2.** Alteration of the interface between eIF2α-D1 and C-terminal helix of uS7 in the open versus closed conformations of the py48S PIC. (A, B) Depiction of the py48S PIC (PDB 3J81) showing uS7/Rps5 (gold), mRNA (orange), Met-tRNAi (green), eIF2α (purple). For clarity, other ribosomal proteins, eIF2β, eIF2γ, eIF1, eIF1A and putative eIF5 densities are not shown. uS7 residues previously implicated in promoting AUG recognition (*Visweswaraiah et al., 2015*) are shown in blue or red with stick side-chains. (C) Overlay of py48S-open (PDB 3JAQ) and py48S-closed (PDB 3JAP)

*Figure 2 continued on next page*

*Figure 2 continued*

revealing remodeling of the interface between eIF2α-D1 (purple or dark blue-closed complex; magenta or orange-open complex) and C-terminal helix of uS7 (beige-closed, yellow-open). Residues making contacts that appear to be favored in the open or closed state are shown with stick side-chains, using dotted lines to indicate the favored interactions.

recognition of the AUG codon of eIF1 (*SUI1*) mRNA, present in poor context, and increased the probability that scanning PICs bypass, or 'leaky scan' past, the AUG codon of upstream open reading frame 1 (uORF1) in *GCN4* mRNA. The Glu-144 substitution (E144R) also dramatically destabilized TC binding to PICs reconstituted with an AUG or UUG start codon in mRNA, with a stronger effect for UUG (*Visweswaraiah et al., 2015*). Together, these findings implicated Arg-225 and amino acids in the uS7 β-hairpin, particularly Glu-144, in stabilizing the $P_{IN}$ conformation of the PIC, and revealed a requirement for these residues in preventing selection of near-cognate (UUG) or AUG start codons in poor context in vivo (*Visweswaraiah et al., 2015*).

The uS7 substitutions with the greatest effects on start codon recognition are located in the upper portion of the β-hairpin (E144R) or at the very C-terminus (R225K), distant from the context nucleotides in mRNA; whereas substitutions of residues in the loop of the β-hairpin, including R148E, which contacts the mRNA directly (*Figure 2B*), had relatively weaker phenotypes (*Visweswaraiah et al., 2015*). Thus, it was unclear what molecular interactions in the PIC are perturbed by the E144R and R225K substitutions. Interestingly, both E144 and R225 interact with other uS7 residues located in the C-terminal helix, which in turn interacts extensively with eIF2α-D1 (*Hussain et al., 2014*) (*Figure 2B*). As eIF2α-D1 also interacts with the anticodon stem-loop of tRNA$_i$ (*Figure 2B*), we considered that the strong defects in start codon recognition conferred by E144R and R225K might result from an altered orientation of the uS7 C-terminal helix that perturbs its interaction with eIF2α-D1 in a way that indirectly destabilizes TC binding in the $P_{IN}$ state (*Visweswaraiah et al., 2015*). Because it was unknown whether the interface between eIF2α-D1 and the uS7 C-terminal helix is important for start codon recognition, we set out here to determine whether uS7 substitutions predicted to perturb this interface would alter the accuracy of start codon recognition in vivo.

Recent cryo-EM analysis has revealed a partial yeast PIC exhibiting a more open configuration of the mRNA binding cleft and P site (py48S-open) compared to both the previous py48S structure (*Hussain et al., 2014*) and a similar complex also containing eIF3 (py48S-closed) (*Llácer et al., 2015*). The py48S-open complex exhibits an upward movement of the 40S head from the body that both widens the mRNA binding cleft and opens the entry channel latch, and evokes a widened P site lacking interactions between Met-tRNA$_i$ and the 40S body found in py48S-closed. These features of py48S-open seem well-suited to the scanning of successive triplets entering the P site for complementarity to Met-tRNA$_i$ with TC anchored in a relatively unstable conformation (*Llácer et al., 2015*). During the transition from py48S-open to py48S-closed, eIF2α-D1 rotates slightly to avoid a clash with the 40S body, which alters the interface between eIF2α-D1 and the C-terminal helix of uS7. Certain contacts appear to be enhanced in the open conformation (*Figure 2C*; D77-R219 and D84-S223) and thus might be expected to promote continued scanning through UUG or 'poor-context' AUG codons and thereby increase initiation accuracy. A third contact (*Figure 2C*; Y82-D215) is favored in the closed conformation and might have the opposite function of enabling recognition of suboptimal initiation sites by promoting the highly stable $P_{IN}$ conformation of TC binding to the closed complex. Thus, to examine the importance of the eIF2α-D1/uS7 interface in start codon recognition, we chose to perturb these predicted contacts that appear to be favored in one PIC conformation or the other and determine their effects on initiation at poor initiation codons in vivo and the stability of TC binding to reconstituted PICs in vitro. Our results support the physiological importance of the differential contacts between uS7 and eIF2α-D1 in the py48S-open and py48S-closed structures in modulating the transition to the $P_{IN}$ conformation by the scanning PIC and, hence, the accuracy of start codon selection.

## Results

### Substitutions of uS7 Asp-215 increase discrimination against suboptimal initiation codons in vivo

The cryo-EM structure of the py48S complex reveals two sites of interaction between eIF2α-D1 and uS7: (i) loops in eIF2α-D1 and the uS7 $\beta$-hairpin, both in proximity to the $-3$ nucleotide in mRNA; and (ii) the C-terminal helix of uS7 and residues in the $\beta$-barrel structure of eIF2α-D1 (*Figure 2A–B*). Comparison of the py48S-open and –closed structures (*Llácer et al., 2015*) suggests that interactions of uS7 residues R219 and S223 with eIF2α D77 and D84, respectively, are more favored in the open conformation, whereas uS7 D215 interaction with eIF2α Y82 is more favored in the closed state (*Figure 2C*). Thus, disrupting these interactions might alter the fidelity of start codon selection in different ways. In particular, disrupting the uS7-D215/eIF2α-Y82 contact favored in the closed state (*Figure 3A*) might increase discrimination against near-cognate UUG or poor-context AUG codons by shifting the system to the open/$P_{OUT}$ conformation conducive to scanning (*Figure 1*). To test this hypothesis, we introduced Leu, Ala or Phe substitutions of uS7 D215 by mutagenesis of an *RPS5* allele under its own promoter on a low-copy plasmid, and examined the phenotypes in a yeast strain harboring wild-type (WT) chromosomal *RPS5* under a galactose-inducible promoter ($P_{GAL1}$-*RPS5$^+$*). Despite strong sequence conservation of uS7 D215 in diverse eukaryotes (*Visweswaraiah et al., 2015*), none of the mutations substantially reduced the ability of plasmid-borne *RPS5* to rescue WT cell growth following a shift to glucose medium to repress $P_{GAL1}$-*RPS5* expression (*Figure 3B*, Glu).

To determine whether the D215 substitutions increase discrimination against non-AUG codons, we asked whether they suppress the elevated initiation at the UUG start codon of mutant *his4–301* mRNA, which lacks an AUG start codon, conferred by a dominant Sui$^-$ mutation (*SUI5*) in the gene encoding eIF5 (*TIF5*). As expected (*Huang et al., 1997*), *SUI5* overcomes the histidine auxotrophy conferred by *his4–301* in the *RPS5$^+$* strain (*Figure 3C*, -His, rows 1–2); and, importantly, this His$^+$/Sui$^-$ phenotype is diminished by all three D215 substitutions (*Figure 3C*, -His, rows 3–5). The *D215L* allele also suppresses the slow-growth phenotype conferred by *SUI5* on histidine-supplemented (+His) medium (*Figure 3C*, +His, rows 1 and 3), a known attribute of eIF1 Ssu$^-$ mutations described previously (*Martin-Marcos et al., 2011*). The D215 substitutions also mitigate the elevated expression of a *HIS4-lacZ* reporter containing a UUG start codon, relative to a matched AUG reporter, conferred by a dominant Sui$^-$ mutation in the eIF2$\beta$ gene (*SUI3–2*; *Huang et al., 1997*) (*Figure 3D*), thus confirming their Ssu$^-$ phenotypes. These results suggest that replacing the acidic side chain of D215 with the hydrophobic side chains of Ala, Leu, or Phe perturbs the uS7/eIF2α-D1 interface in a way that impedes inappropriate transition to the closed/$P_{IN}$ state at UUG start codons conferred by Sui$^-$ variants of eIF5 or eIF2$\beta$.

As *D215L* appears to have the strongest Ssu$^-$ phenotype among the alleles tested, we examined its effect on 40S subunit biogenesis or stability, and bulk translation in vivo. Consistent with its WT growth, the *D215L* mutant showed no reduction in the ratio of polysomes to 80S monosomes (P/M ratio) versus WT, suggesting a nearly WT rate of bulk protein synthesis (*Figure 3E*). *D215L* cells also display a nearly WT ratio of total 40S to 60S subunits, measured under conditions that dissociate 80S ribosomes into free subunits (*Figure 3F*), indicating little or no effect of D215L on 40S biogenesis or stability. Thus, the enhanced initiation accuracy conferred by *D215L* appears to reflect an increased propensity of the mutant 43S PIC to bypass a near-cognate start codon during scanning rather than a reduction in 40S abundance.

In addition to reducing initiation from near-cognate UUG codons, certain Ssu$^-$ mutations in eIF1 and eIF1A reduce initiation from AUG codons in poor context. As such, they exacerbate the effects of the native, suboptimal context of the AUG codon of *SUI1* mRNA and decrease expression of the encoded eIF1 protein (*Martin-Marcos et al., 2011*). All three D215 Ssu$^-$ substitutions similarly reduced eIF1 expression (*Figure 4A*) and, consistently, reduced expression of a *SUI1-lacZ* reporter bearing the native, suboptimal context at the nucleotides preceding the AUG codon ($^{-3}$CGU$^{-1}$), while modestly increasing expression of a modified *SUI1$_{opt}$-lacZ* reporter with optimized context ($^{-3}$AAA$^{-1}$) (*Figure 4B*). As expected, expression of the *SUI1$_{opt}$-lacZ* reporter is 2-fold higher than that of *SUI1-lacZ* in *RPS5$^+$*cells (*Martin-Marcos et al., 2011*), whereas the *SUI1$_{opt}$-lacZ/SUI-lacZ* expression ratio is elevated to between 3- and 4-fold in the D215 mutants (*Figure 4B*). Thus, the D215 substitutions exacerbate the effect of suboptimal context and decrease AUG recognition on native *SUI1* mRNA. The reduction in eIF1 abundance implies that the D215 substitutions overcome

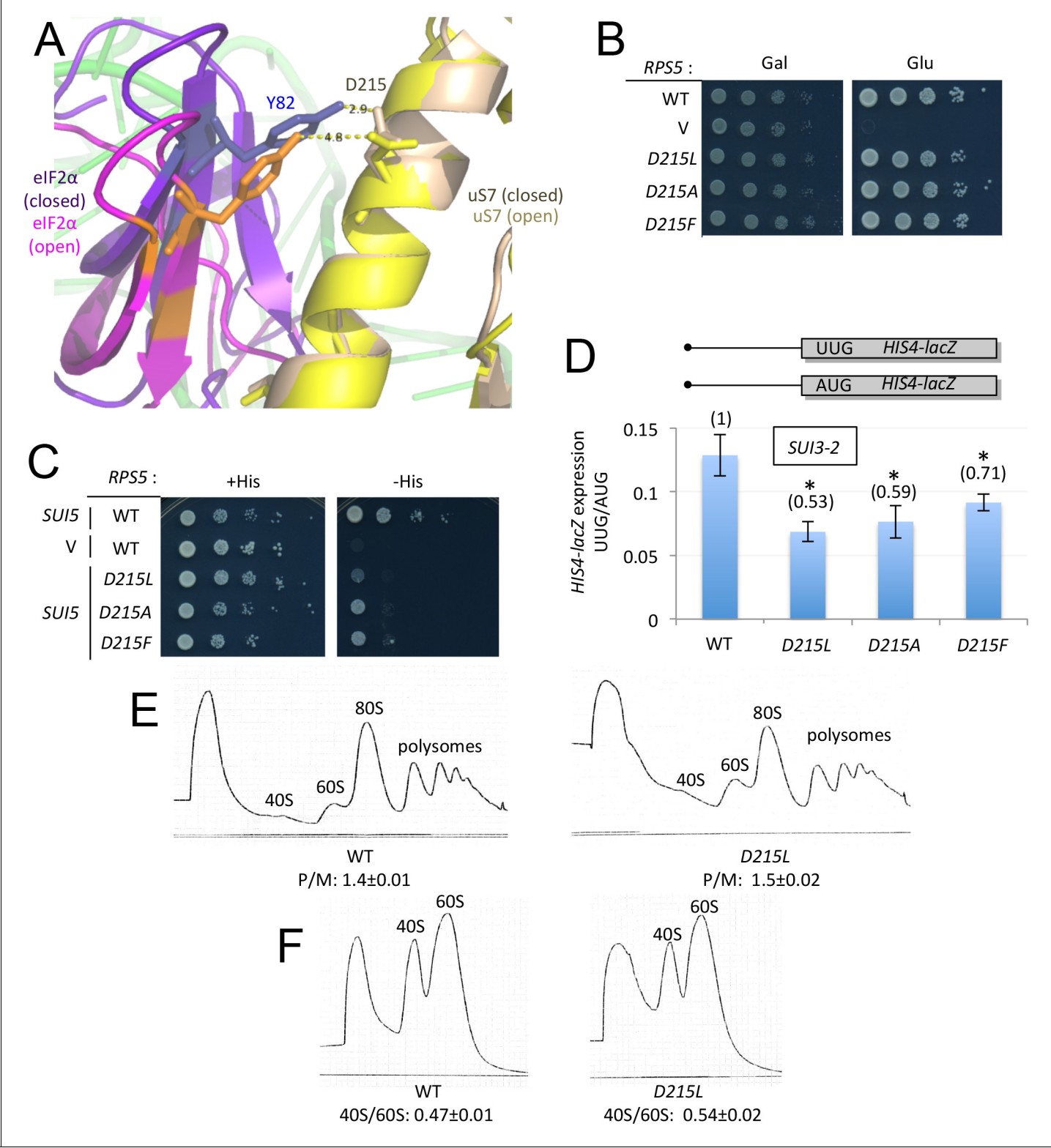

**Figure 3.** uS7-D215 substitutions increase discrimination against UUG start codons in vivo. (A) Overlay of py48S-open and py48S-closed as in *Figure 2C*, showing that uS7-D215/eIF2α-Y82 interaction is favored in the closed complex (dark blue/beige sticks). (B) 10-fold serial dilutions of transformants of *pGAL1-RPS5 his4–301* strain (JVY07) with the indicated plasmid-borne *RPS5* alleles, or empty vector (V) were spotted on SC_Gal-Leu (Gal) or SC-Leu (Glu) and incubated at 30°C for 2–3 days. (C) 10-fold serial dilutions of JVY07 transformants with the indicated *RPS5* alleles and *SUI5* plasmid p4281, or empty vector (V) were spotted on SD+Ura+His (+His) or SD+Ura (−His) and incubated at 30°C for 3d and 5d, respectively. (D) JVY07

*Figure 3 continued on next page*

**Figure 3 continued**

transformants with the indicated *RPS5* alleles, *SUI3–2* plasmid p4280, and *HIS4-lacZ* reporters with AUG or UUG start codons (plasmids p367 and p391, respectively) were cultured in SD+His at 30°C to an $A_{600}$ of ~1 and $\beta$-galactosidase specific activities were measured in WCEs in units of nanomoles of o-nitrophenyl-$\beta$-D-galactopyranoside (ONPG) cleaved per min per mg of total protein. Ratios of mean expression of the UUG and AUG reporters calculated from four biological and two technical replicates are plotted with error bars (indicating S.E.M.s). *p<0.05 (**E**) WT and JVY76 (*rps5-D215L*) were cultured in SC-Leu at 30°C to $A_{600}$ of ~1, and cycloheximide was added prior to harvesting. WCEs were separated by sucrose density gradient centrifugation and scanned at 254 nm to yield the tracings shown. Mean polysome/monosome ratios (and S.E.M.s) from three biological replicates are indicated. (**F**) Similar to (**E**) but cultures were not treated with cycloheximide and lysed in buffers without $MgCl_2$ to allow separation of dissociated 40S and 60S ribosomal subunits. Mean 40S/60S ratios (and S.E.M.s) from three biological replicates are indicated.

The following source data is available for figure 3:

**Source data 1.** Effects of Rps5-D215 substitutions on *HIS4-lacZ* UUG:AUG expression ratios and polysome:monosome ratios.

the autoregulation of eIF1 expression, wherein low eIF1 levels suppress poor context at the *SUI1* AUG codon to restore eIF1 abundance (*Ivanov et al., 2010*; *Martin-Marcos et al., 2011*). Hence, these substitutions confer a pronounced defect in recognition of the *SUI1* AUG codon that prevails even at low cellular concentrations of eIF1 that favor recognition of this suboptimal initiation site.

We asked next whether the D215L Ssu⁻ substitution can decrease recognition of the AUG codon of an upstream ORF (uORF) by assaying a *GCN4-lacZ* reporter harboring a modified version of uORF1, elongated to overlap the *GCN4* ORF (el.uORF1), as the sole uORF in the mRNA leader. With the native, optimum context of the uORF1 AUG ($^{-3}$AAA$^{-1}$), virtually all scanning ribosomes translate el.uORF1 and subsequent reinitiation at the *GCN4-lacZ* ORF is nearly non-existent, such that *GCN4-lacZ* translation of this reporter is very low (*Grant et al., 1994*) (*Figure 4C*, col. 1, row 1). In agreement with previous work (*Visweswaraiah et al., 2015*), replacing optimum context with the weaker context $^{-3}$UAA$^{-1}$ at uAUG-1 increases leaky scanning of el.uORF1 and elevates *GCN4-lacZ* expression ~8 fold; an even greater ~30 fold increase in *GCN4-lacZ* expression is conferred by the extremely poor context $^{-3}$UUU$^{-1}$; and elimination of uAUG-1 increases *GCN4-lacZ* expression by ~100 fold (*Figure 4C*, col. 1, rows 1–4). Based on these results, the percentages of scanning ribosomes that either translate el.uORF1 or leaky-scan uAUG-1 and translate *GCN4-lacZ* instead can be calculated (*Figure 4C*, cols. 3 and 5), revealing that about 99%, 93%, and 71% of scanning ribosomes recognize uAUG-1 in optimum, weak, or poor context, respectively, in WT cells (*Figure 4C* col. 5, rows 1–3). Note that while leaky-scanning to *GCN4-lacZ* increases by ~30 fold on replacing optimum with poor context, this entails only a ~30% reduction in el.uORF1 translation (*Figure 4C*, col. 5), as virtually no leaky-scanning (1%) occurs at uAUG-1 in optimum context (*Figure 4C*, col. 3).

The uS7 D215L substitution increases leaky scanning of el.uORF1 and elevates *GCN4-lacZ* expression between ~2.5 and 4-fold for the different reporters containing uAUG-1, while having relatively little effect on the uORF-less reporter (*Figure 4C*, col. 1 vs 2 and col. 3 vs. 4, rows 1–4). Comparing the percentages of scanning ribosomes that initiate at uAUG-1 in *D215L* and WT cells, as calculated above, reveals that *D215L* reduces initiation at uAUG-1 by ~17% and~41% for weak and poor contexts, respectively, but only by ~1% for optimum context (*Figure 4C*, cf. cols. 5 and 6). Thus, *D215L* preferentially discriminates against uAUG-1 in weak or poor context, in accordance with its relatively greater effect on initiation at the *SUI1-lacZ* AUG in native, poor context (*Figure 4B*).

Previously, we showed that Ssu⁻ substitutions E144R and R225K in the $\beta$-hairpin loop of uS7 exhibit the same phenotypes described above for D215L, reducing initiation at the native *SUI1* AUG codon and increasing leaky scanning of *GCN4* uAUG-1 in optimum, weak, or poor context (*Visweswaraiah et al., 2015*). To determine whether E144R/R225K preferentially discriminate against uAUG-1 in poor context, we calculated their effects on the fraction of scanning ribosomes that initiate at el.uORF1 for each context of uAUG-1 in the manner shown in *Figure 4C* for D215L. As shown in *Figure 4—figure supplement 1*, R225K and E144R both resemble D215L in preferentially decreasing el.uORF1 translation for weak and poor context versus optimum context. In fact, E144R essentially eliminates recognition of uAUG-1 in poor context, while reducing it only slightly for optimum context (*Figure 4* Fig. sup., cf. cols. 7 and 9). These findings support the possibility that uS7 R225K/E144R confer hyperaccurcy phenotypes by indirectly perturbing the uS7/eIF2α-I interface in the manner altered directly by the D215 substitutions.

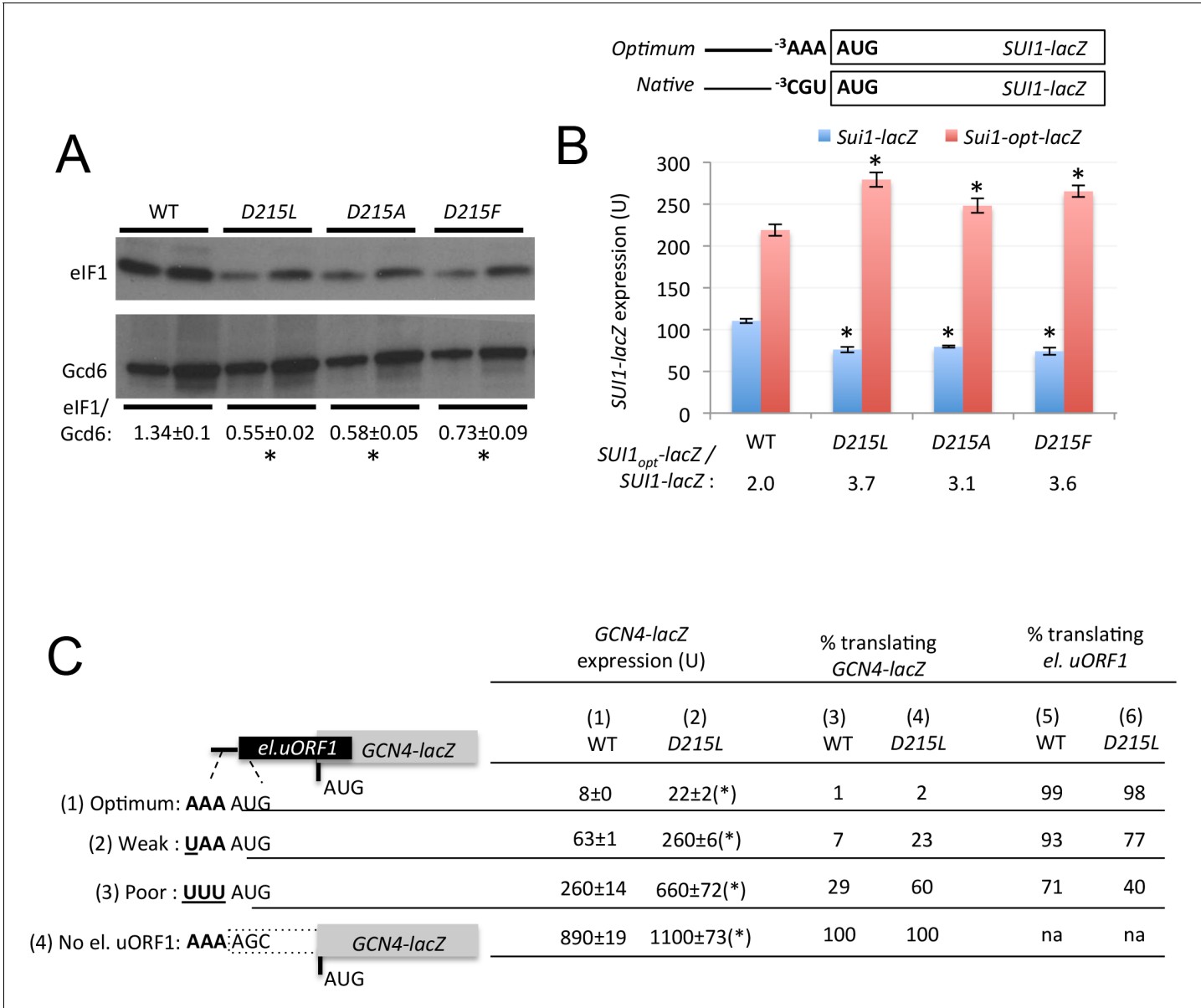

**Figure 4.** uS7 substitution D215L discriminates against AUG start codons in poor context. (**A**) WCEs of strains from *Figure 3B* subjected to Western analysis using antibodies against eIF1 or Gcd6 (as loading control). Two amounts of each extract differing by a factor of two were loaded in successive lanes. Signal intensities from four biological replicates were quantified and mean eIF1/Gcd6 ratios are listed below the blot with S.E.Ms. *p<0.05 (**B**) Strains from *Figure 3B* also harboring *SUI1-lacZ* (pPMB24) or *SUI1-opt-lacZ* (pPMB25) reporters, containing native or optimum context at positions −1 to −3, were assayed for β-galactosidase activities as in *Figure 3D*. Mean expression levels and S.E.M.s from four biological and two technical replicates are plotted, and ratio of mean expression levels of *SUI1-lacZ* reporters with optimized context to native context are listed below the histogram. *p<0.05 (**C**) β-galactosidase activities measured in WCEs of WT and uS7-D215L transformants harboring the *el.uORF1 GCN4-lacZ* reporters pC3502, pC4466, or pC3503 containing, respectively, the depicted optimum, weak, or poor context of uAUG-1; or the uORF-less *GCN4-lacZ* reporter pC3505 with mutated uAUG-1. Mean expression values with S.E.M.s were determined from three biological and two technical replicates and listed in columns 1 and 2. Cols. 3–4 gives the percentage of ribosomes translating the *GCN4-lacZ* ORF in the different constructs, calculated as a percentage of the *GCN4-lacZ* activity observed for the 'no el. uORF1' construct measured for the relevant construct shown in cols. 1–2. Cols. 5–6 gives the percentage of ribosomes translating el.uORF, calculated as 100% minus the percentage translating the *GCN4-lacZ* ORF shown in cols. 3–4. (*), p<0.05.

The following source data and figure supplement are available for figure 4:

**Source data 1.** Source data for *Figure 4* and *Figure 4—figure supplement 1*.
**Figure supplement 1.** uS7 β-hairpin Ssu⁻ substitutions R225K and E144R discriminate against AUG start codons in poor context.

## Ssu⁻ uS7 substitution D215L destabilizes the P$_{IN}$ conformation of the 48S PIC in vitro

The multiple defects in start codon recognition conferred by *rps5-D215L* suggest that it destabilizes the P$_{IN}$ state of the 48S PIC. We tested this hypothesis by analyzing the effects of the uS7 D215L substitution on TC binding to the 40S subunit in the yeast reconstituted translation system. We began by measuring the affinity of WT TC, assembled with [$^{35}$S]-Met-tRNA$_i$, for 40S subunits harboring mutant or WT uS7 in the presence of saturating eIF1, eIF1A and a model unstructured mRNA containing an AUG start codon (mRNA(AUG)), using native gel electrophoresis to separate 40S-bound and unbound fractions of labeled TC. The 40S subunits were purified from *rps5Δ::kanMX* deletion strains harboring either plasmid-borne *rps5-D215L* or *RPS5⁺* as the only source of uS7. The reconstituted 40S•eIF1•eIF1A•mRNA•TC complexes will be referred to as partial 43S•mRNA complexes owing to the absence of eIF3 and eIF5, which are dispensable for PIC assembly using these model mRNAs (*Algire et al., 2002*). Reactions conducted with increasing concentrations of 40S subunits revealed that the partial 43S•mRNA(AUG) complexes containing D215L or WT 40S subunits have K$_d$ values of ≤1 nM (*Figure 5A and D*). While this assay is not sensitive enough to detect decreases in TC affinity unless they exceed two-orders of magnitude (*Kolitz et al., 2009*), the results indicate that stable partial 43S•mRNA(AUG) complexes can be assembled with D215L mutant 40S subunits. In the absence of mRNA, the affinities for TC were also similar between partial 43S PICs assembled with mutant or WT 40S subunits (*Figure 5B and D*).

We next determined the rate constants for TC dissociation from 43S·mRNA complexes using mRNAs harboring AUG or UUG start codons. To measure the TC off-rate (k$_{off}$), partial 43S•mRNA complexes were formed as above using TC assembled with [$^{35}$S]-Met-tRNA$_i$, and the amount of [$^{35}$S]-Met-tRNA$_i$ remaining in the slowly-migrating PIC was measured at different times after adding a chase of excess unlabeled TC. To mimic the situation in vivo where *D215L* suppressed the Sui⁻ phenotype of *SUI3–2* (*Figure 3D*), we measured the k$_{off}$ using eIF2 harboring the eIF2β substitution (S264Y) encoded by *SUI3–2*. Consistent with our previous results (*Martin-Marcos et al., 2014*), in reactions with WT 40S subunits, TC dissociates from AUG complexes very little over the time course of the experiment, yielding a rate constant of only 0.06 h$^{-1}$ (*Figure 5C*; summarized in *Figure 5E*). TC dissociation from WT PICs assembled on an otherwise identical mRNA containing a UUG start codon is also relatively slow (k$_{off}$ = 0.10 h$^{-1}$), owing to the stabilization of complexes at UUG codons conferred by the *SUI3–2* mutation in eIF2β (*Figure 5C and E*). Importantly, the TC dissociation rates for partial 43S•mRNA complexes assembled with *D215L* 40S subunits was increased ~3 fold for mRNA(AUG) and ~8 fold for mRNA(UUG) compared to the k$_{off}$ values of the corresponding WT complexes (*Figure 5C and E*). These findings provide biochemical evidence that *D215L* destabilizes P$_{IN}$ at both AUG and UUG start codons with a relatively stronger effect on the near-cognate triplet, overriding the opposing effect of *SUI3–2* of enhancing the stability of the UUG complex. These in vitro findings are in accordance with the in vivo effects of *D215L* of reducing recognition of the *SUI1* AUG and *GCN4* uAUG-1 start codons, and suppressing the elevated UUG:AUG initiation ratio on *his4–301* mRNA conferred by *SUI3–2*.

## Substitutions of uS7 residues Arg-219 and Ser-223 decrease discrimination against suboptimal initiation codons in vivo

As noted above, comparing the structures of py48S-open and -closed (*Llácer et al., 2015*) suggests that interactions of uS7 residues R219 and S223 with eIF2α-D1 residues D77 and D84, respectively, are both favored in the open complex (*Figure 2C and 6A*), such that disrupting these interactions might decrease discrimination against near-cognate UUG or poor-context AUG start codons by enhancing transition to the closed/P$_{IN}$ conformation required for start codon selection (*Figure 1*). Supporting this hypothesis, Ala and Asp substitutions of R219 conferred strong increases in the UUG:AUG initiation ratio of *HIS4-lacZ* mRNA (*Figure 6B*), indicating Sui⁻ phenotypes. The *R219D* mutation also conferred weak growth on –His medium, despite producing slow-growth (Slg⁻) on +His medium (*Figure 6C*, row 5), indicating elevated initiation at the UUG start codon of *his4–301* mRNA. The His⁺ phenotype of *R219D* was exacerbated by overexpressing eIF5 from a high-copy *TIF5* plasmid, which also conferred a His⁺/Sui⁻ phenotype in *R219A* cells (*Figure 6C*, cf. hc*TIF5* and vector (V) rows). It is known that eIF5 overexpression intensifies UUG initiation in Sui⁻ mutants by promoting eIF1 dissociation and TC binding in the P$_{IN}$ state (*Nanda et al., 2009*). The R219H

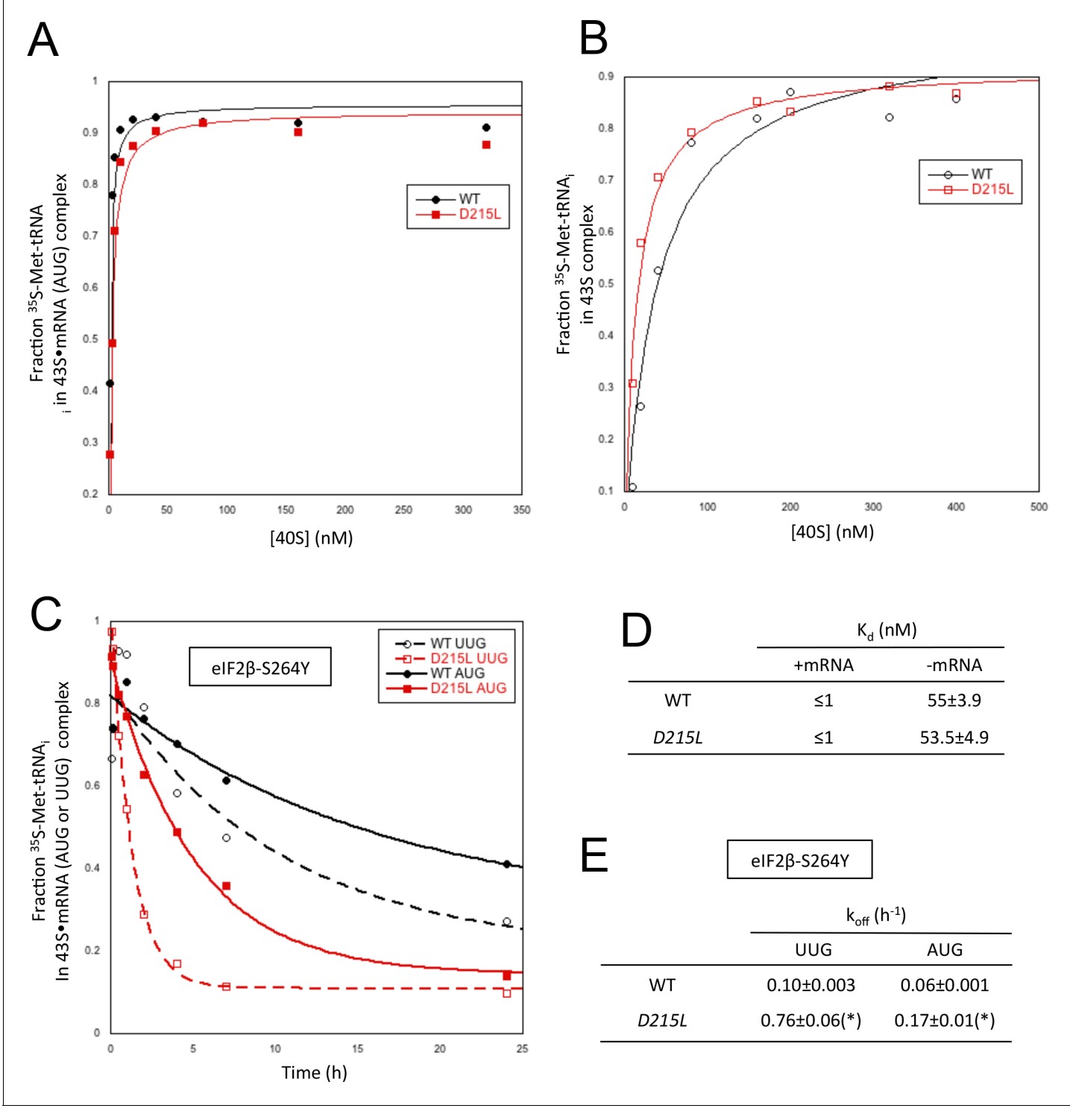

**Figure 5.** uS7 substitution D215L destabilizes $P_{IN}$ in vitro preferentially at UUG start codons. (**A, B**) Determination of $K_d$ values for TC with [$^{35}$S]-Met-tRNA$_i$ binding to 40S·eIF1·eIF1A complexes assembled with WT or D215L mutant 40S subunits and either mRNA (AUG) (**A**) or without mRNA (**B**). (**C**) Analysis of TC dissociation kinetics from 43S·mRNA complexes assembled with WT or D215L mutant 40S subunits and mRNA(AUG) or mRNA(UUG), conducted using the eIF2β-S264Y Sui⁻ variant of eIF2. Representative curves selected from three independent experiments are shown. (**D, E**) $K_d$ and $k_{off}$ values with S.E.M.s from three independent experiments determined in (**A–C**). (*), p<0.05.

The following source data is available for figure 5:

*Figure 5 continued on next page*

*Figure 5 continued*

**Source data 1.** Effects of Rps5-D215L on TC affinity for partial 43S and 43S·mRNA complexes, and rate of TC dissociation from partial 43S·mRNA complexes reconstituted with the eIF2β-S264Y variant of eIF2.

substitution, by contrast, confers only a modest increase in UUG:AUG initiation (*Figure 6B*) and does not display a His⁺ phenotype even with eIF5 overexpression (*Figure 6C*, rows 7–8).

Similar to Sui⁻ mutations in eIF1, eIF1A, and eIF2β (*Martin-Marcos et al., 2011*), the uS7 R219D and R219A substitutions reduce discrimination against the native, poor context of the *SUI1* AUG codon and evoke increased eIF1 expression (*Figure 6D*). Consistently, they also confer increased expression of the *SUI1-lacZ* reporter with native, poor context. They also increase expression of *SUI1opt-lacZ* (with optimal context), but to a lesser degree, and thereby diminish the *SUI1opt-lacZ/ SUI1-lacZ* expression ratio (*Figure 6E*). In accordance with its lack of Sui⁻ phenotype, the *R219H* mutation has little or no effect on eIF1 expression (*Figure 6D*) or the *SUI1opt-lacZ/ SUI1-lacZ* expression ratio (*Figure 6E*). Assaying expression of the *el.uORF1-GCN4-lacZ* reporters revealed that *R219D* confers decreased leaky scanning of uAUG-1 and attendant reduced translation of the downstream *GCN4-lacZ* ORF (*Figure 6F*, cf. cols. 1–2). Calculating the fraction of scanning ribosomes that translate el.uORF1 indicates a substantial increase in recognition of uAUG-1 in poor context, a smaller increase with uAUG-1 in weak context, and a negligible change with uAUG-1 in optimal context (*Figure 6F*, cf. columns 5–6). Thus, it appears that eliminating the basic side-chain of Arg-219 (R219A) or substituting it with an acidic side-chain (R219D) confers moderate or severe disruptions, respectively, of the uS7/eIF2α-D1 interface to facilitate inappropriate transition to the closed/P_IN state at both UUG codons and AUGs in poor-context. The relatively stronger phenotype of the Asp substitution of R219 might reflect electrostatic repulsion with D77 in eIF2α-D1 (*Figure 6A*). The Slg⁻ phenotype of *rps5-R219D* (*Figure 6C*, +His, row 5) is associated with diminished polysome assembly, indicated by a reduced P/M ratio (*Figure 6—figure supplement 1A*); which does not arise from a reduction in 40S subunit abundance (*Figure 6—figure supplement 1B*).

Interaction of uS7 Ser-223 with eIF2α-D1 residue Asp-84 also appears to be favored in the open complex (*Figure 7A*). Similar to our findings for the R219D/A substitutions, replacing Ser-223 with Ala, Arg, Asp, or Phe, evokes increased UUG initiation, with S223D conferring the greatest increase in the UUG:AUG *HIS4-lacZ* initiation ratio (*Figure 7D*). Consistently, *S223D* also suppresses the His⁻ phenotype of *his4–301* despite a strong Slg⁻ defect on +His medium (*Figure 7B*). Furthermore, S223D was the only substitution of Ser-223 that both increased eIF1 expression (*Figure 7C*) and decreased the *SUI1opt-lacZ/ SUI1-lacZ* expression ratio (*Figure 7E*), signifying reduced discrimination against the native (poor) context of the *SUI1* AUG codon. However, we found that *S223D* did not significantly increase recognition of uAUG-1 of el.uORF1 in poor or weak context to reduce expression of the corresponding *el.uORF1-GCN4-lacZ* reporters, indicating a narrower effect of reducing discrimination against poor context than observed for the R219D substitution (*Figure 6D–F*).

In accordance with its strong Slg⁻ phenotype, *S223D* confers a marked reduction in polysomes (*Figure 7G*) without appreciably altering 40S subunit abundance (*Figure 7H*), indicating a defect in bulk translation initiation. Numerous Sui⁻ mutations affecting eIF1 (*Cheung et al., 2007*; *Nanda et al., 2009*; *Martin-Marcos et al., 2013*), eIF1A (*Fekete et al., 2005*; *Saini et al., 2010*), and tRNAi^Met were shown to reduce the rate of TC loading on 40S PICs, presumably by destabilizing the P_OUT conformation of TC binding, conferring constitutive derepression of *GCN4* mRNA (the Gcd⁻ phenotype). A slower rate of TC recruitment allows 40S subunits that have translated uORF1 and resumed scanning to bypass the start codons of inhibitory uORFs 2–4 before rebinding TC, and then reinitiate further downstream at the *GCN4* AUG codon instead. Interestingly, *S223D* also produces a strong Gcd⁻ phenotype, depressing *GCN4-lacZ* expression by ~5 fold (*Figure 7F*). Thus, it appears that introducing an acidic side chain at the position of S223 perturbs the uS7/eIF2α-D1 interface in the open complex to destabilize the P_OUT mode of TC binding and confer the Gcd⁻ phenotype, facilitate inappropriate transition to the closed/P_IN state at UUG codons or the *SUI1* AUG codon, and produce a general reduction in the rate of translation initiation. The fact that the Asp substitution produces a much stronger phenotype than the other three substitutions of S223 might arise from the introduction of electrostatic repulsion with Asp-84 in eIF2α-D1 (*Figure 7A*).

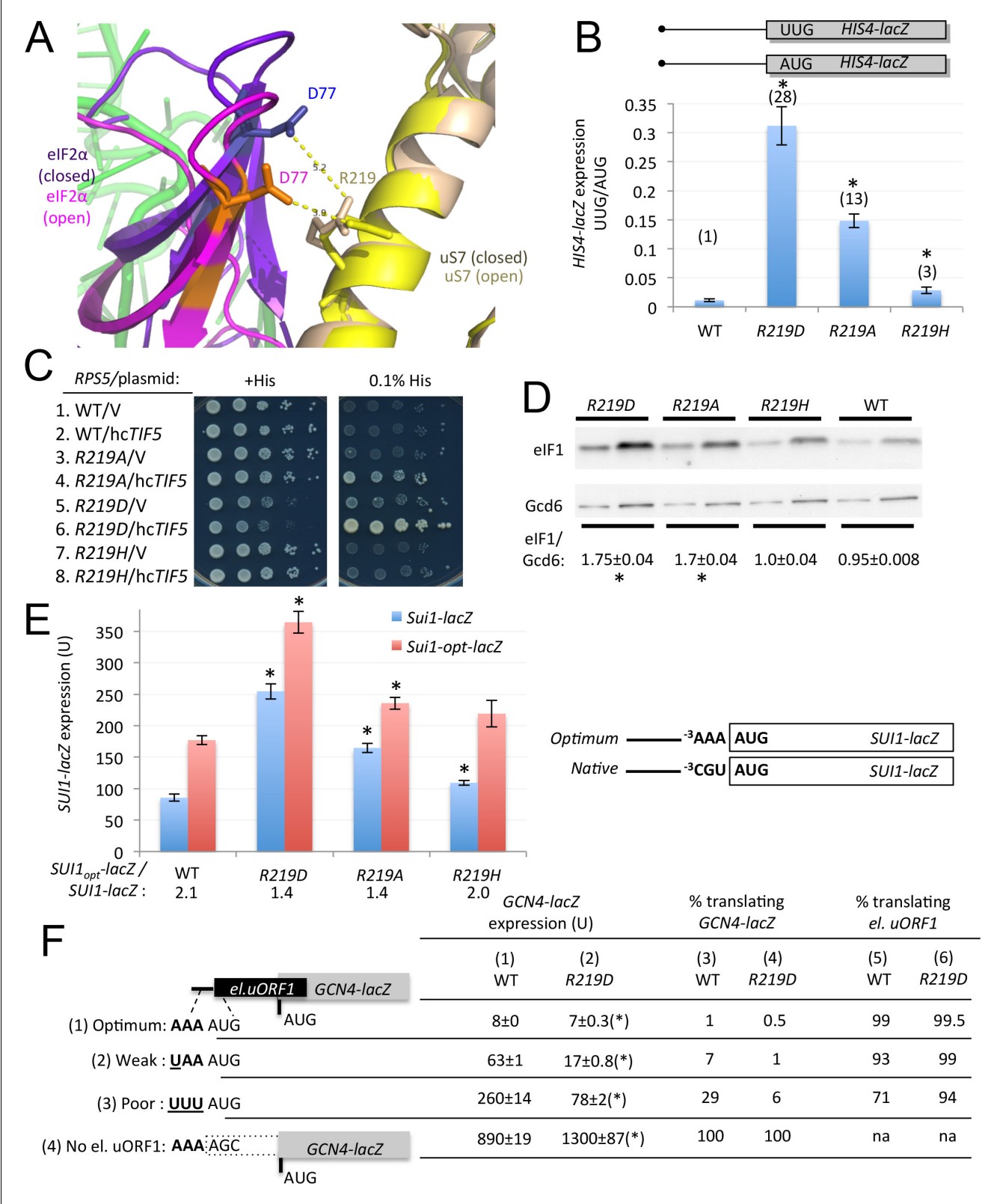

**Figure 6.** uS7 substitution R219D increases initiation at UUG codons and AUG codons in poor context. (**A**) Overlay of py48S-open and py48S-closed showing uS7-R219/eIF2α-D77 interaction favored in the open complex (orange/yellow sticks). (**B**) Ratio of expression of *HIS4-lacZ* reporters with AUG or UUG start codons in transformants of JVY07 determined as in *Figure 3D*. Mean ratios and S.E.M.s calculated from four biological and two technical replicates. *p<0.05 (**C**) 10-fold serial dilutions of JVY07 transformants harboring the indicated *RPS5* alleles and high-copy *TIF5* plasmid p4438 or empty

*Figure 6 continued on next page*

*Figure 6 continued*

vector (V) spotted on SD+His+Ura (+His) or SD+Ura+0.0003 mM His (0.1% of usual His supplement; -His) and incubated at 30°C for 3d. (**D**) WCEs of three biological replicate strains from (**B**) subjected to Western analysis of eIF1 expression, as in *Figure 4A*. *p<0.05 (**E**) Expression of *SUI1-lacZ* or *SUI1-opt-lacZ* reporters in transformants of strains from (**B**), determined as in *Figure 4B*. Mean expression levels and S.E.M.s were calculated from four biological and two technical replicates. *p<0.05 (**F**) Expression of *el.uORF1 GCN4-lacZ* reporters in transformants of the WT or *rps5-219D* strains from (**B**), analyzed as in *Figure 4C*. (*), p<0.05.

The following source data and figure supplement are available for figure 6:

**Source data 1.** Source data for *Figure 6* and *Figure 6—figure supplement 1*.
**Figure supplement 1.** uS7 substitution R219D decreases bulk translation initiation but does not derepress translation of *GCN4* mRNA.

## Sui $^-$ uS7 substitution S223D promotes the $P_{IN}$ conformation of the 48S PIC in vitro

Because the S223D substitution confers the strongest Sui$^-$ and Gcd$^-$ phenotypes among the uS7 substitutions that appear to specifically disrupt the open/$P_{OUT}$ conformation of the PIC, we purified mutant 40S subunits harboring this uS7 variant and measured the affinity and rate constants for TC binding in vitro. The S223D substitution had no significant effect on the $K_d$ values for TC binding to partial 43S•mRNA(AUG) complexes, or partial 43S complexes lacking mRNA, but appeared to reduce the end-point for TC binding to 43S complexes lacking mRNA (*Figure 8A–B*). As this failure to achieve a WT end-point at saturating concentrations of 40S subunits likely indicates dissociation of PICs during gel electrophoresis (*Kapp et al., 2006*; *Kolitz et al., 2009*), the results indicate destabilization of the $P_{OUT}$ mode of TC binding to partial 43S complexes containing uS7-S223D.

Interestingly, measuring the rate of TC dissociation from partial 43S·mRNA complexes revealed that S223D reduces the rate of TC dissociation from complexes harboring AUG or UUG start codons, essentially eliminating measurable dissociation from the AUG complex and decreasing the $k_{off}$ for the UUG complex by ~5 fold compared to the WT value (*Figure 8C–D*). We also measured rates of TC binding to these complexes ($k_{on}$) by mixing labeled TC [$^{35}$S]-Met-tRNA$_i$ with different concentrations of 40S subunits and saturating eIF1, eIF1A and mRNA(AUG) or mRNA(UUG), removing aliquots at different time points and terminating reactions with excess unlabeled TC. The amount of labeled TC incorporated into PICs as a function of time yields the pseudo-first-order rate constant ($k_{obs}$) for each 40S concentration, and the slope of the plot of $k_{obs}$ versus 40S concentration yields the second-order rate constant ($k_{on}$) (*Kolitz et al., 2009*). As shown in *Figure 8E–F*, S223D increased the $k_{on}$ values for AUG and UUG PICs by ~2 fold and 4-fold, respectively. As the rate constant measured in these experiments is thought to be a composite of the rate of initial binding of TC to the PIC in the $P_{OUT}$ state followed by transition from $P_{OUT}$ to $P_{IN}$ (*Kolitz et al., 2009*), the increase in $k_{on}$ conferred by S223D could indicate acceleration of one or both steps. However, considering that *S223D* confers a Gcd$^-$ phenotype in vivo (*Figure 7D*), signifying a reduced rate of TC loading to 40S subunits (*Hinnebusch, 2011*), and also appears to destabilize the $P_{OUT}$ state of TC binding to 43S complexes lacking mRNA (end-point defect in *Figure 8A–B*), it seems probable that the increased $k_{on}$ results from accelerating the transition from the $P_{OUT}$ to $P_{IN}$ states of TC binding to the PIC. This interpretation is supported by our finding that $k_{on}$ is increased more substantially for UUG versus AUG complexes (*Figure 8F*), whereas the initial loading of TC on the PIC should be independent of the start codon (*Kolitz et al., 2009*). In fact, the actual acceleration of $P_{OUT}$ to $P_{IN}$ conversion conferred by *S223D* is likely to be substantially greater than the 2–to 4-fold increases in measured $k_{on}$ values, as this effect would be offset by the decreased rates of TC binding in the $P_{OUT}$ state predicted by the Gcd$^-$ phenotype of *S223D* in vivo. Thus, taken together, the results in *Figure 8* provide biochemical evidence that *S223D* enhances conversion from the $P_{OUT}$ state to the highly stable $P_{IN}$ conformation at both AUG and UUG start codons, in accordance with the effects of this mutation in vivo of increasing recognition of the poor-context *SUI1* AUG codon and elevating near-cognate UUG initiation on *his4–301* mRNA during ribosomal scanning.

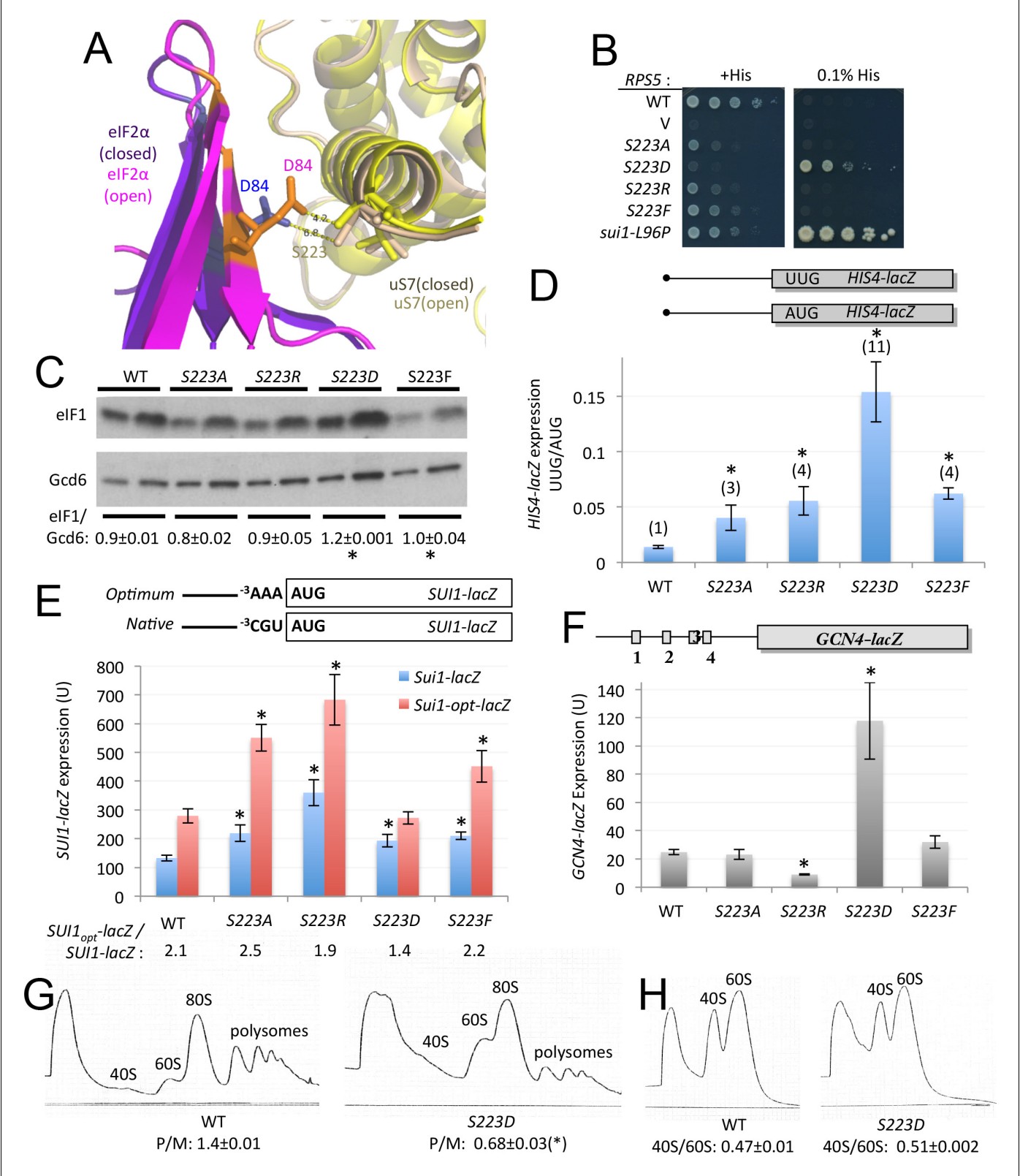

**Figure 7.** uS7 S223 substitutions decrease initiation fidelity in vivo. (**A**) Overlay of py48S-open and py48S-closed complexes showing uS7-S223/eIF2α-D84 interaction favored in the open complex (orange/yellow sticks). (**B**) Dilutions of JVY07 transformed with the indicated *RPS5* alleles and *sui1-L96P* strain H4564 spotted on SD+His+Ura+Trp (+His) or SD+Ura+Trp+0.0003 mM His (-His) and incubated at 30℃ for 3 and 5 d, respectively. (**C**) WCEs of three biological replicate strains from (**B**) subjected to Western analysis of eIF1 expression as in *Figure 4A*. *p<0.05 (**D**) Ratio of expression of *HIS4-lacZ*

*Figure 7 continued on next page*

*Figure 7 continued*

reporters with AUG or UUG start codons in transformants of strains from (**B**), determined as described in *Figure 3D*. Mean ratios and S.E.M.s calculated from four biological and two technical replicates. *p<0.05 (**E**) Expression of *SUI1-lacZ* or *SUI1-opt-lacZ* reporters in transformants of strains from (**B**), determined as in *Figure 4B*. Mean expression levels and S.E.M.s calculated from four biological and two technical replicates. *p<0.05 (**F**) Expression of WT *GCN4-lacZ* in transformants of strains from (**B**), determined as in *Figure 3D*, with mean expression levels and S.E.M.s calculated from four biological and two technical replicates. *p<0.05 (**G–H**) Polysome to monosome ratios (**G**) and 40S/60S ratios (**H**) in WT and *rps5-S223D* strains from (**B**), determined as in *Figure 3E–F* with mean ratios and S.E.M.s calculated from three biological replicates. (*), p<0.05.

The following source data is available for figure 7:

**Source data 1.** Effects of Rps5-S223 substitutions on eIF1 expression, *HIS4-lacZ* UUG:AUG expression ratios, *SUI1$_{opt}$-lacZ: SUI1$_{nat}$-lacZ* expression ratios, *GCN4-lacZ* expression, and polysome:monosome ratios.

## Discussion

We previously implicated the $\beta$-hairpin of uS7 in achieving efficient and accurate start codon recognition (*Vishwesvaraiah et al., 2015*), but the molecular interactions involved in these functions were unclear. Here, using a combination of genetics and biochemistry, we obtained strong evidence that uS7 influences start codon recognition through direct interactions with domain 1 of eIF2α. Structural analyses of reconstituted yeast PICs revealed that eIF2α-D1 interacts with both the anticodon stem of tRNA$_i$, mRNA residues immediately upstream of the AUG codon, and the C-terminal helix of uS7, and suggested that the uS7/eIF2α-D1 interface is remodeled during the transition from the open conformation, thought to be conducive to scanning, to the closed state required for start codon recognition (*Llácer et al., 2015*). We made targeted substitutions of uS7 residues whose contacts with specific amino acids in eIF2α-D1 appear to be favored in the open or closed conformation and thus might contribute differentially to the stabilities of these two states. As such, altering these contacts should have opposing effects on the probability of switching from the open, scanning conformation to the closed state at suboptimal start codons, including near-cognate UUG triplets and AUGs in poor surrounding context. Fulfilling these predictions would not only implicate the uS7/eIF2α-D1 interface in modulating start codon recognition, but also provide evidence that the different PIC conformations revealed by the structural studies represent physiological intermediates of the initiation pathway.

In accordance with the predictions based on the PIC structures (*Llácer et al., 2015*), we found that substitutions perturbing the uS7-D215/eIF2α-Y82 interaction favored in the closed state reduce initiation at UUG codons in cells harboring Sui$^-$ mutations in eIF2β or eIF5 (that aberrantly elevate UUG initiation), and also decrease recognition of AUGs in poor context in otherwise WT cells, including the native, suboptimal start codon of the eIF1 gene (*SUI1*), and uAUG-1 of *GCN4* uORF1 when it resides in weak or poor context. The potent uS7 substitution D215L was shown to destabilize the P$_{IN}$ state of TC binding to the PIC in vitro, using the *SUI3–2* variant of eIF2β to assemble TC, increasing the dissociation rate of TC (k$_{off}$) with a relatively stronger effect at UUG versus AUG start codons. These findings suggest that the uS7-D215/eIF2α-Y82 contact preferentially stabilizes the P$_{IN}$ state (*Figure 1*), and that perturbing this interaction disproportionately discriminates against suboptimal initiation sites whose P$_{IN}$ conformations are inherently less stable and thus hyperdependent on the uS7/eIF2α interface present in the closed conformation for their efficient utilization in cells. The D215L substitution resembles the E144R substitution in the uS7 $\beta$-hairpin loop in increasing discrimination against poor initiation codons and preferentially destabilizing the P$_{IN}$ state at UUG codons (*Vishwesvaraiah et al., 2015*), supporting the notion that altering the $\beta$-hairpin loop confers hyperaccurate initiation by indirectly perturbing the uS7/eIF2α-I interface in the closed PIC.

Remarkably, uS7 substitutions altering two other contacts that seem to be favored in the open conformation, uS7-R219/eIF2α-D77 and uS7-S223/eIF2α-D84, had the opposite effects on the system, compared to uS7-D215L, of enhancing utilization of a UUG start codon, the suboptimal *SUI1* AUG codon, and (at least for R219A/D substitutions) *GCN4* uAUG-1 in weak or poor context. Moreover, the potent uS7 substitution S223D also had the opposite effect in vitro of stabilizing the P$_{IN}$ state of TC binding to the 48S PIC, decreasing k$_{off}$ at UUG codons. Interestingly, uS7-S223D also accelerates formation of the closed/P$_{IN}$ complex, thus increasing k$_{on}$; and the relatively stronger increase in k$_{on}$ observed for the UUG versus AUG complex suggests that the P$_{OUT}$ to P$_{IN}$ transition,

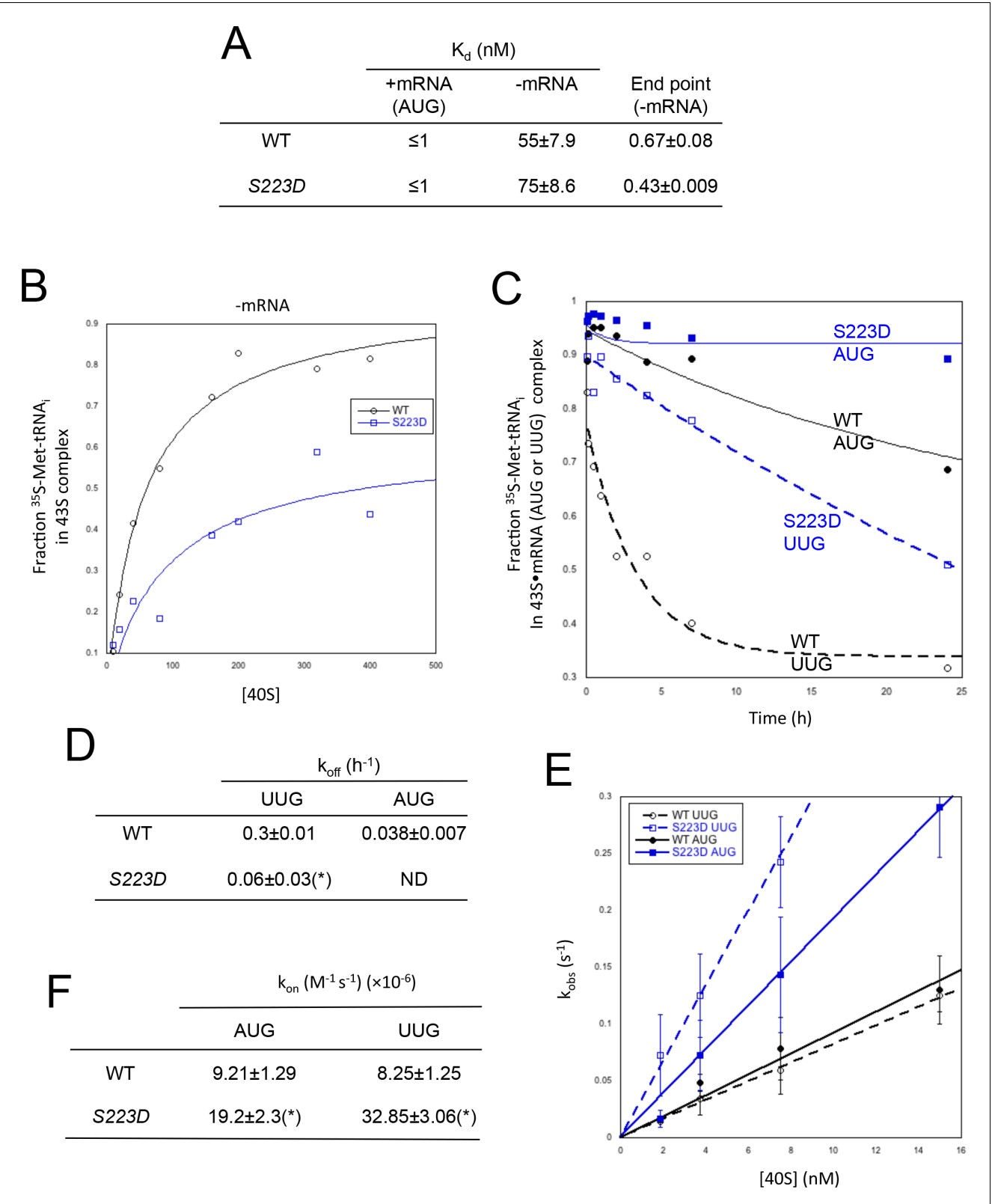

**Figure 8.** uS7 substitution S223D promotes $P_{IN}$ at UUG codons. (**A–B**) Mean $K_d$ and end-point values with S.E.M.s for binding of TC assembled with [$^{35}$S]-Met-tRNA$_i$ to 40S·eIF1·eIF1A complexes reconstituted with WT or Rps5-S223D mutant 40S subunits and either mRNA (AUG) or without mRNA, determined from three independent experiments. A representative experiment is shown in (**B**). (**C–D**) Analysis of TC dissociation kinetics for 43S·mRNA complexes assembled with WT or Rps5-S223D mutant 40S subunits and either mRNA(AUG) or mRNA(UUG). A representative curve selected from three

*Figure 8 continued on next page*

*Figure 8 continued*

independent experiments is shown in (C), and mean $k_{off}$ values with S.E.M.s are given in (D). (*), p<0.05 (E–F) Determination of $k_{on}$ values for TC binding to 40S·eIF1·eIF1A complexes from plots of observed rate constants ($k_{obs}$) vs 40S concentration for WT or Rps5-S223D mutant 40S subunits and mRNA (AUG or UUG) shown in (E) with S.E.M.s of $k_{obs}$ values for at least three independent experiments at each 40S concentration. Mean $k_{on}$ values with S.E.M.s calculated from three independent experiments are given in (F). (*), p<0.05.

The following source data is available for figure 8:

**Source data 1.** Effects of Rps5-S223D on TC affinity for partial 43S and 43S·mRNA complexes, and rates of TC association and dissociation from partial 43S·mRNA complexes.

rather than initial loading of TC to PIC, is accelerated by S223D. In fact, based on the Gcd⁻ phenotype conferred by *S223D* in vivo, the initial loading of TC in the $P_{OUT}$ configuration appears to be impaired by S223D. Together, these results suggest that uS7-S223D enhances the transition from the relatively less stable $P_{OUT}$ conformation to the more stable $P_{IN}$ state of TC binding by destabilizing the $P_{OUT}$ conformation, which decreases the rate of TC recruitment during reinitiation events on *GCN4* mRNA (to evoke the Gcd⁻ phenotype) and also enhances selection of suboptimal initiation codons during scanning, including the native eIF1 start codon, *GCN4* uAUG-1 in poor context, and UUG start codons (the Sui⁻ phenotype).

The dual Sui⁻/Gcd⁻ phenotypes of *rps5-S223D* have been observed for numerous mutations affecting various eIFs (*Hinnebusch, 2011*), including substitutions in eIF1 that weaken its binding to the 40S subunit (*Martin-Marcos et al., 2013*). Because eIF1 accelerates TC loading in the $P_{OUT}$ state but physically impedes the $P_{OUT}$ to $P_{IN}$ transition by clashing with tRNA$_i$ in the $P_{IN}$ conformation (*Passmore et al., 2007*; *Rabl et al., 2011*; *Hussain et al., 2014*), the reduced 40S association of these eIF1 variants reduces the rate of TC binding (Gcd⁻ phenotype) and simultaneously enhances rearrangement to $P_{IN}$ at UUG codons (Sui⁻ phenotype) (*Martin-Marcos et al., 2013*). In the case of *rps5-S223D*, both the Gcd⁻ and Sui⁻ phenotypes likely result from weakening direct interaction of uS7 with eIF2α-D1 in the TC specifically in the $P_{OUT}$ state, which both delays TC loading and increases the probability of $P_{OUT}$ to $P_{IN}$ transition. Unlike *S223D,* we found that the strong Sui⁻ allele *rps5-R219D* does not confer a Gcd⁻ phenotype (*Figure 6—figure supplement 1C*), which might indicate that the uS7-R219/eIF2α-D77 interaction in the open conformation is relatively more important for impeding the $P_{OUT}$ to $P_{IN}$ transition than for accelerating TC loading in the $P_{OUT}$ state.

In summary, our results provide strong evidence that the interface between the C-terminal helix of uS7 and eIF2α-D1 participates in recruitment of TC in the $P_{OUT}$ conformation and modulates the transition between the open and closed conformations of the PIC during the scanning process to establish the wild-type level of discrimination against near-cognate UUG triplets and AUG codons in poor context as initiation sites. The opposing consequences on initiation accuracy in vivo and the rates of TC dissociation from reconstituted partial PICs in vitro conferred by the uS7 substitutions D215L and S223D provides evidence that the distinct conformations of the uS7/eIF2α-D1 interface seen in the py48S-open and py48S-closed structures described by *Llácer et al. (2015)*, which are differentially perturbed by these two uS7 substitutions, are physiologically relevant to the mechanism of scanning and accurate start codon selection.

## Materials and methods

### Plasmids and yeast strains

Yeast strains used in this study are listed in *Table 1*. Derivatives of JVY07 harboring low copy (lc) *LEU2* plasmids containing *RPS5⁺* (pJV09) or mutant *RPS5* alleles (pJV67-pJV84 listed in *Table 2*) were generated by transformation to yield strains JVY31-JVY94, respectively, listed in *Table 1*. Haploid strains JVY98 and JVY99 harboring *rps5-D215L* and *rps5-S223D,* respectively as the only source of uS7 were generated by plasmid shuffling as described previously (*Visweswaraiah et al., 2015*).

Plasmids used in this study are listed in *Table 2*. *RPS5* fragments were amplified by fusion PCR to introduce the desired site-directed mutations, using pJV09 as template DNA. The mutagenized fragments were digested with BglII and NdeI and inserted between the same two restriction sites in

**Table 1.** Yeast strains employed in this study.

| Strain | Genotype | Source or reference |
|---|---|---|
| H4564 | MATa ura3–52 trp1Δ−63 leu2–3,112 his4-301(ACG) sui1Δ::hisG pPMB03 (sc LEU2 sui1-L96P) | (Martin-Marcos et al., 2011) |
| JVY07 | MATa ura3–52 trp1Δ−63 leu2–3,112 his4-301(ACG) kanMX6:$P_{GAL1}$-RPS5 | (Visweswaraiah et al., 2015) |
| JVY31 | MATa ura3–52 trp1Δ−63 leu2–3,112 his4-301(ACG) kanMX6:$P_{GAL1}$-RPS5 pJV09 (lc LEU2 RPS5) | (Visweswaraiah et al., 2015) |
| JVY76 | MATa ura3–52 trp1Δ−63 leu2–3,112 his4-301(ACG) kanMX6:$P_{GAL1}$-RPS5 pJV09 (lc LEU2 rps5-D215L) | This study |
| JVY77 | MATa ura3–52 trp1Δ−63 leu2–3,112 his4-301(ACG) kanMX6:$P_{GAL1}$-RPS5 pJV09 (lc LEU2 rps5-D215A) | This study |
| JVY78 | MATa ura3–52 trp1Δ−63 leu2–3,112 his4-301(ACG) kanMX6:$P_{GAL1}$-RPS5 pJV09 (lc LEU2 rps5-D215F) | This study |
| JVY85 | MATa ura3–52 trp1Δ−63 leu2–3,112 his4-301(ACG) kanMX6:$P_{GAL1}$-RPS5 pJV09 (lc LEU2 rps5-R219D) | This study |
| JVY86 | MATa ura3–52 trp1Δ−63 leu2–3,112 his4-301(ACG) kanMX6:$P_{GAL1}$-RPS5 pJV09 (lc LEU2 rps5-R219A) | This study |
| JVY87 | MATa ura3–52 trp1Δ−63 leu2–3,112 his4-301(ACG) kanMX6:$P_{GAL1}$-RPS5 pJV09 (lc LEU2 rps5-R219H) | This study |
| JVY91 | MATa ura3–52 trp1Δ−63 leu2–3,112 his4-301(ACG) kanMX6:$P_{GAL1}$-RPS5 pJV09 (lc LEU2 rps5-S223A) | (Visweswaraiah et al., 2015) |
| JVY92 | MATa ura3–52 trp1Δ−63 leu2–3,112 his4-301(ACG) kanMX6:$P_{GAL1}$-RPS5 pJV09 (lc LEU2 rps5-S223D) | This study |
| JVY93 | MATa ura3–52 trp1Δ−63 leu2–3,112 his4-301(ACG) kanMX6:$P_{GAL1}$-RPS5 pJV09 (lc LEU2 rps5-S223R) | This study |
| JVY94 | MATa ura3–52 trp1Δ−63 leu2–3,112 his4-301(ACG) kanMX6:$P_{GAL1}$-RPS5 pJV09 (lc LEU2 rps5-S223F) | This study |
| JVY11 | MATα ura3-Δ0 leu2-Δ0 his3Δ−1 lys2-Δ0 MET15 rps5Δ::kanMX pJV38 (lc URA3 RPS5) | (Visweswaraiah et al., 2015) |
| JVY15 | MATα ura3-Δ0 leu2-Δ0 his3Δ−1 lys2-Δ0 MET15 rps5Δ::kanMX pJV13 (lc LEU2 RPS5) | (Visweswaraiah et al., 2015) |
| JVY98 | MATα ura3-Δ0 leu2-Δ0 his3Δ−1 lys2-Δ0 MET15 rps5Δ::kanMX pJV09 (lc LEU2 rps5-D215L) | This study |
| JVY99 | MATα ura3-Δ0 leu2-Δ0 his3Δ−1 lys2-Δ0 MET15 rps5Δ::kanMX pJV35 (lc LEU2 rps5-S223D) | This study |

pJV09, to produce pJV67-pJV84. All constructs were verified by DNA sequencing of 1 kb from the inserted BglII site beyond the NdeI restriction site, covering the entire *RPS5* ORF.

## Plate assays

Saturated overnight cultures were subject to ten-fold serial dilutions, and 5 µl of each dilution was transferred to plates. If necessary, galactose was used as sole carbon source instead of glucose to induce the expression of genes under the galactose-inducible promoter. The plates were incubated at 30°C until colonies were visible. The assays were conducted on two biological replicates (independent transformants) and representative plates from one replicate are presented.

## Biochemical analyses of yeast cells

Assays of $\beta$-galactosidase activity in whole-cell extracts (WCEs) were performed as described previously (Moehle and Hinnebusch, 1991). All $\beta$-galactosidase activity assays were performed with two technical replicates using the same extracts. To determine the UUG to AUG initiation ratio, matched *HIS4-lacZ* reporters with UUG or AUG as start codon were used. The sequence context of the start codon for both AUG and UUG *HIS4-lacZ* reporters is: 5'-AUA(AUG/UUG)G-3'. Four biological replicates (independent transformants) with two technical replicates were employed for all UUG to AUG ratio measurements, and the S.E.M.s for the ratios were calculated as (X/Y) ($\sqrt{[(SE_x/x)^2+(SE_y/y)^2]}$, where X, $SE_x$, and x are the mean, standard error of the mean, and highest values for the UUG reporter, respectively; and Y, $SE_y$, and y are the corresponding values for the AUG reporter). For Western analysis, WCEs from three biological replicates (independent cultures) were prepared by trichloroacetic acid extraction as described (Reid and Schatz, 1982), and immunoblot analysis was conducted as described previously (Martin-Marcos et al., 2011) with antibodies against eIF1 (Valásek et al., 2004) or Gcd6 (Bushman et al., 1993). Enhanced chemiluminescence (Amersham, Pittsburgh, PA) was used to visualize immune complexes, and signal intensities were quantified by densitometry using NIH ImageJ software.

**Table 2.** Plasmids employed in this study.

| Plasmid | Description[*] | Source or reference |
|---|---|---|
| pJV09 | lc *LEU2 RPS5* with BglII site engineered in pJV01 | (*Visweswaraiah et al., 2015*) |
| pJV38 | lc *URA3 RPS5* with BglII site engineered in pRS316 | (*Visweswaraiah et al., 2015*) |
| pJV67 | lc *LEU2 rps5-D215L* | This study |
| pJV68 | lc *LEU2 rps5-D215A* | This study |
| pJV69 | lc *LEU2 rps5-D215F* | This study |
| pJV76 | lc *LEU2 rps5-R219D* | This study |
| pJV77 | lc *LEU2 rps5-R219A* | This study |
| pJV78 | lc *LEU2 rps5-R219H* | This study |
| pJV33 | lc *LEU2 rps5-S223A* | (*Visweswaraiah et al., 2015*) |
| pJV82 | lc *LEU2 rps5-S223R* | This study |
| pJV83 | lc *LEU2 rps5-S223D* | This study |
| pJV84 | lc *LEU2 rps5-S223F* | This study |
| p367 | sc *URA3 HIS4(ATG)-lacZ* | (*Donahue and Cigan, 1988*) |
| p391 | sc *URA3 HIS4(TTG)-lacZ* | (*Donahue and Cigan, 1988*) |
| p180 | sc *URA3 GCN4-lacZ* in YCp50 | (*Hinnebusch, 1985*) |
| p4280/YCpSUI3-S264Y-W | sc *TRP1 SUI3-S264Y* in YCplac22 | (*Valásek et al., 2004*) |
| p4281/YCpTIF5-G31R-W | sc *TRP1 TIF5-G31R* in YCplac22 | (*Valásek et al., 2004*) |
| p4438/YEplacTIF5-W | hc *TRP1 TIF5* in YEplac112 | Christie Fekete |
| pPMB24 | sc *URA3 SUI1-lacZ* | (*Martin-Marcos et al., 2011*) |
| pPMB25 | sc *URA3 SUI1-opt-lacZ* | (*Martin-Marcos et al., 2011*) |
| pC3502 | sc *URA3* $^{-3}$AAA$^{-1}$ el.uORF1 *GCN4-lacZ* in YCp50 | (*Visweswaraiah et al., 2015*) |
| pC4466 | sc *URA3* $^{-3}$UAA$^{-1}$ el.uORF1 *GCN4-lacZ* in YCp50 | (*Visweswaraiah et al., 2015*) |
| pC3503 | sc *URA3* $^{-3}$UUU$^{-1}$ el.uORF1 *GCN4-lacZ* in YCp50 | (*Visweswaraiah et al., 2015*) |
| pC3505 | sc *URA3* el.uORF1-less *GCN4-lacZ* in YCp50 | (*Visweswaraiah et al., 2015*) |

[*]lc, low copy number; sc, single copy; hc, high copy.

## Polysome and ribosomal subunit profiling

For polysome analysis, yeast strains were grown in SC-Leu at 30°C to $A_{600}$, ~1. Cycloheximide was added (50 µg/ml) 5 min prior to harvesting, and WCE was prepared in breaking buffer (20 mM Tris-HCl, pH 7.5, 50 mM KCl, 10 mM MgCl$_2$, 1 mM dithiothreitol, 5 mM NaF, 1 mM phenylmethylsulfonyl fluoride, 1 Complete EDTA-free Protease Inhibitor Tablet (Roche. Indianapolis, IN)/50 mL buffer). 15 $A_{260}$ units of WCE from at least three biological replicates were separated by velocity sedimentation on a 4.5% to 45% sucrose gradient by centrifugation at 39,000 rpm for 3 hr in an SW41Ti rotor (Beckman Coulter, Indianapolis, IN). Gradient fractions were scanned at 254 nm to visualize ribosomal species. For analysis of total 40S to 60S subunit ratios, yeast strains were grown in SC-Leu at 30°C to $A_{600}$, ~1 and harvested without cycloheximide treatment. WCE was prepared in breaking buffer lacking MgCl$_2$ (20 mM Tris HCl, pH 7.5, 50 mM NaCl, 1 mM dithiothreitol, 1 mM phenylmethylsulfonyl fluoride, 200 ug/ml heparin, 1 Complete EDTA-free Protease Inhibitor Tablet (Roche)/50 mL buffer). 15 $A_{260}$ units of WCE from three biological replicates (independent cultures) were separated by velocity sedimentation as described for polysome profiling.

## Biochemical analysis in the reconstituted yeast translation system

Initiation factors eIF1A and eIF1 were expressed in *E. coli* and purified using the IMPACT system (New England Biolabls, Ipswich, MA), and His$_6$-tagged eIF2 was overexpressed in yeast and purified as described (*Acker et al., 2007*). WT and mutant 40S subunits were purified from yeast as described previously (*Acker et al., 2007*). Model mRNAs with the sequences 5'-GGAA[UC]$_7$UAUG

[CU]$_{10}$C-3' and 5'-GGAA[UC]$_7$UUUG[CU]$_{10}$C-3' were purchased from Thermo Scientific. Yeast tRNA$_i$-$^{Met}$was synthesized from a hammerhead fusion template using T7 RNA polymerase and charged with [$^{35}$S]-methionine or unlabeled methionine as previously described (*Acker et al., 2007*). K$_d$ values of TC (assembled with [$^{35}$S]-Met-tRNA$_i$) and 40S•eIF1•eIF1A•mRNA PICs, and rate constants of TC association/dissociation for the same PICs, were determined by gel shift assays as described previously (*Kolitz et al., 2009*) with the minor modifications described in (*Visweswaraiah et al., 2015*).

## Statistical analysis

Unpaired student's t-test was performed to compare wild type and mutant mean values and the change was considered significant if the two-tailed P value was < 0.05.

## Acknowledgements

We thank Fan Zhang for assistance in performing certain experiments. We thank Laura Marler and Anil Thakur for valuable discussions, Thomas Dever, Jon Lorsch and members of their laboratories and our own for helpful advice. This work was supported in part by the Intramural Program of the National Institutes of Health.

## Additional information

### Competing interests

AGH: Reviewing editor, *eLife*. The other author declares that no competing interests exist.

### Funding

| Funder | Grant reference number | Author |
| --- | --- | --- |
| National Institutes of Health | Intramural Program HD001004 | Alan G Hinnebusch |

The funders had no role in study design, data collection and interpretation, or the decision to submit the work for publication.

### Author contributions

JV, Conceptualization, Formal analysis, Validation, Investigation, Methodology, Writing—original draft, Writing—review and editing; AGH, Conceptualization, Formal analysis, Supervision, Writing—original draft, Writing—review and editing

### Author ORCIDs

Alan G Hinnebusch, http://orcid.org/0000-0002-1627-8395

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
