## [Decision Letter]

Thank you for submitting your article "Interactions between eIF2α and 40S exit channel protein uS7/Rps5 modulate start codon recognition in vivo" for consideration by *eLife*. Your article has been reviewed by two peer reviewers, and the evaluation has been overseen by a Reviewing Editor and James Manley as the Senior Editor. The following individual involved in review of your submission has agreed to reveal his identity: Christopher S Fraser (Reviewer #3).

The reviewers have discussed the reviews with one another and the Reviewing Editor has drafted this decision to help you prepare a revised submission.

The authors probe the interface between RpS5/uS7 and eIF2α through mutational analysis for its role in PIC open and closed states. The presented data support the conclusions and the problem addressed is of fundamental significance.

1) It is possible that some mutations uS7 cause partial misfolding or altered domain orientation in a way that perturbs interaction with eIF2α. In this case, the observed phenotype of some of the mutants would not be due to disruption of uS7: eIF2α interaction. For the R219D and S223D mutants it should be possible to rescuing the phenotypes by compensatory D77R and D84S mutations, respectively, on eIF2α. The generation of these obligate dimers would help support the physiological importance of the differential contacts between uS7 and eIF2α-D1 in the py48S-open and py48S-closed structure.

2) The authors place mutants of uS7 D215 by mutagenesis of an RPS5 allele under its own promoter on a low-copy plasmid, and examined the phenotypes in a yeast strain harboring wild-type (WT) chromosomal RPS5 under a galactose-inducible promoter (PGAL1-RPS5+). How much mutant protein (D215 and others) vs. wild-type protein is present in the 40S ribosomes following the shift to glucose medium at the time when these experiments are performed?

---

## [Author Response]

*The authors probe the interface between RpS5/uS7 and eIF2α through mutational analysis for its role in PIC open and closed states. The presented data support the conclusions and the problem addressed is of fundamental significance.*

1) It is possible that some mutations uS7 cause partial misfolding or altered domain orientation in a way that perturbs interaction with eIF2α. In this case, the observed phenotype of some of the mutants would not be due to disruption of uS7: eIF2α interaction. For the R219D and S223D mutants it should be possible to rescuing the phenotypes by compensatory D77R and D84S mutations, respectively, on eIF2α. The generation of these obligate dimers would help support the physiological importance of the differential contacts between uS7 and eIF2α-D1 in the py48S-open and py48S-closed structure.

The reviewer asks that we attempt to suppress the strong Sui- phenotype of the R219D substitution, which is thought to eliminate the Rps5-R219/eIF2α-D77 salt bridge, by making a compensatory eIF2α-D77R substitution to restore the salt bridge. Similarly, we were requested to look for suppression of the Rps5-S223D substitution with an eIF2α-D84S substitution. This an interesting suggestion; however, testing it would require the construction of a new strain in which mutant alleles of both RPS5 and the gene encoding eIF2α (SUI2) can be combined in a double mutant, which is not a trivial matter. We began the construction of this strain, but have not been able to complete it yet. In addition, there are several reasons to think that the outcome of the proposed eIF2α substitutions could be more complicated than imagined. Considering the proposed eIF2α-D77R substitution, we note first that eIF2α-D84 is within 8Å of eIF2α-D77 (refer to Figure 2), and given the longer length of the Arg vs. Asp side-chain, a new salt bridge could be formed between the proposed eIF2α-D77R substitution and wild-type eIF2α-D84 rather than the intended salt bridge between eIF2α-D77R and Rps5-R219D. This in turn could prevent eIF2α-D84 from interacting with Rps5-S223 in the open complex and thus produce a Sui- phenotype indirectly. Second, eIF2α-D77 is within 4Å of eIF2α-Y82, and the proposed D77R substitution could impair the interaction of eIF2α-Y82 with Rps5-D215 in the closed complex (Figure 2), and thus confer an Ssu^-^ phenotype indirectly. Third, although not highlighted in Figure 2, Rps5-R219 is within 4Å of eIF2α-Y82 exclusively in the open complex, and interaction between these residues might contribute to the stability of the open complex. If so, then our Rps5-R219D substitution might perturb interaction with eIF2α-Y82 as well as with eIF2α-D77, contributing to the strong Sui- phenotype of Rps5-R219D, in a way that could not be suppressed by the proposed eIF2α-D77R substitution.

The outcome of the proposed eIF2α-D84S substitution is also uncertain because, in addition to eIF2α-D84’s interaction with Rps5-S223 in the open complex (shown in Figure 2), D84 might interact with Rps5-R219 in the closed complex (as only 5Å separates their side-chains). As such, the proposed eIF2α-D84S substitution could destabilize both the open and closed states with offsetting effects. It is also relevant that these eIF2α residues are probably located in or near the binding pocket for eIF2B and their substitutions could reduce eIF2 recycling and ternary complex formation, which could complicate their effects on accuracy. These uncertainties in the outcome make us reluctant to invest the considerable time and effort required to generate the appropriate SUI2 mutations, construct the double yeast mutants, and carry out the phenotypic tests to compare the double mutant to each single mutant. Instead, we think it would be more profitable to carry out a random mutagenesis of the SUI2 gene and select for mutations that suppress the Sui- phenotype of the Rps5-R219D mutation. This interesting project would be possible once the aforementioned yeast strain can be constructed, but is beyond the scope of the present work.

Although we cannot dismiss the reviewer’s concern that the Rps5 substitutions cause “partial misfolding or an altered domain orientation in a way that perturbs interaction with eIF2α and not disruption of [specific] uS7: eIF2α interactions”, our finding that the substitutions predicted to preferentially destabilize the open or closed PIC conformation confer the predicted, opposing effects on initiation accuracy in vivo and stability of TC binding to the PIC in vitro argues that the mutations produce quite specific effects on Rps5:eIF2α interaction. It is particularly notable that the Rps5-S223D substitution confers greater than WT stability of TC binding (reduced k_OFF_) (Figure 9C-D), and a greater than WT rate of TC binding in the P_IN_ state exclusively at UUG codons (Figure 9E-F); and it seems difficult to imagine that the non-specific effects of this mutation on Rps5 structure envisioned by the referee would produce mutant 40S subunits that perform better than WT subunits in accessing and maintaining a stable P_IN_ complex.

Nevertheless, given that we have not identified the exact contacts between residues in eIF2α and the three Rps5 residues substituted in our study that regulate initiation accuracy, we altered the title of the paper to read “Interface between 40S exit channel protein uS7/Rps5 and eIF2α modulates start codon recognition in vivo”.

2) The authors place mutants of uS7 D215 by mutagenesis of an RPS5 allele under its own promoter on a low-copy plasmid, and examined the phenotypes in a yeast strain harboring wild-type (WT) chromosomal RPS5 under a galactose-inducible promoter (PGAL1-RPS5+). How much mutant protein (D215 and others) vs. wild-type protein is present in the 40S ribosomes following the shift to glucose medium at the time when these experiments are performed?

To answer this question, we have conducted Western analysis of Rps5 abundance on strains containing the chromosomal pGAL-RPS5 allele and either the WT RPS5 LEU2 plasmid or empty LEU2 vector. We cultured these strains under the exact conditions of our various experiments, including cell spotting assays, polysome profiling and assays of lacZ reporters.

Author response image 1.**DOI:**
http://dx.doi.org/10.7554/eLife.22572.020

As shown in Figure 9, the level of Rps5 in the vector-only strains is reduced to very low levels following growth in Glucose media for the specified incubation used in polysome profiling and reporter assays. While not depleted to the same extent in the inoculum for spotting assays, cells will grow for 10-20 additional generations in glucose when forming colonies from single cells in this assay, which will deplete endogenous Rps5 to a much greater degree. Thus, we are confident that the overwhelming majority of Rps5 present in our glucose-grown cultures is plasmid-encoded. It is not surprising that a small fraction of Rps5 expressed from the pGAL-RPS5 allele remains detectable after prolonged growth on glucose medium. This explains why we did not use this depletion system when purifying the mutant ribosomes needed for biochemical analysis, but rather constructed strains in which chromosomal RPS5 was deleted and the mutant allele was the only source of Rps5 in the cells.